# C-Phycoerythrin Prevents Chronic Kidney Disease-Induced Systemic Arterial Hypertension, Avoiding Oxidative Stress and Vascular Dysfunction in Remanent Functional Kidney

**DOI:** 10.3390/md22080337

**Published:** 2024-07-25

**Authors:** Oscar Iván Florencio-Santiago, Vanesa Blas-Valdivia, José Iván Serrano-Contreras, Placido Rojas-Franco, Gerardo Norberto Escalona-Cardoso, Norma Paniagua-Castro, Margarita Franco-Colin, Edgar Cano-Europa

**Affiliations:** 1Laboratorio de Metabolismo I, Departamento de Fisiología, Escuela Nacional de Ciencias Biológicas, Instituto Politécnico Nacional, Ciudad de México 07738, Mexico; 2Laboratorio de Neurobiología, Departamento de Fisiología, Escuela Nacional de Ciencias Biológicas, Instituto Politécnico Nacional, Ciudad de México 07738, Mexico; 3Department of Metabolism, Digestion and Reproduction, Division of Digestive Diseases, Section of Nutrition, Faculty of Medicine, Hammersmith Campus, Imperial College London, London W12 0NN, UK; 4Laboratorio de Farmacología del Desarrollo, Departamento de Fisiología, Escuela Nacional de Ciencias Biológicas, Instituto Politécnico Nacional, Ciudad de México 07738, Mexico

**Keywords:** C-phycoerythrin, phycobiliprotein, *Phormidium persicinum*, 5/6 nephrectomy, chronic kidney disease, oxidative stress, nephroprotection, antihypertensive, vascular dysfunction

## Abstract

Chronic kidney disease (CKD) is a burden in low- and middle-income countries, and a late diagnosis with systemic arterial hypertension (SAH) is the major complication of CKD. C-phycoerythrin (CPE) is a bioactive compound derived from *Phormidium persicinum* that presents anti-inflammatory and antioxidant effects in vitro and nephroprotective effects in vivo. In the current study, we determine the antihypertensive effect of CPE in a 5/6 nephrectomy-induced CKD model using twenty normotensives male Wistar rats, grouped into four groups (n = 5): sham; sham + CPE; 5/6 nephrectomy (NFx); and NFx + CPE. Treatment started a week post-surgery and continued for five weeks, with weekly hemodynamic evaluations. Following treatment, renal function, oxidative stress, and the expression of vascular dysfunction markers were assessed. The renal function analysis revealed CKD hyperfiltration, and the hemodynamic evaluation showed that SAH developed at the third week. AT_1_R upregulation and AT_2_R downregulation together with Mas1/p-Akt/p-eNOS axis were also observed. CPE treatment mitigated renal damage, preserved renal function, and prevented SAH with the modulation of the vasodilative AT_1_R, AT_2_R, and Mas1/pAKT/peNOS axis. This result reveals that CPE prevented CKD progression to SAH by avoiding oxidative stress and vascular dysfunction in the kidneys.

## 1. Introduction

Chronic kidney disease (CKD) is defined as abnormalities of the kidney structure or function present for almost three months, with alterations in the glomerular filtration rate (GFR) and markers of kidney damage, including albuminuria [1,2]. CKD is a significant burden in low- and middle-income countries. Although Mexico does not have an official CKD registry, in 2017, it was estimated that about 14.5 million people suffered from this illness, with an incidence rate of 394.2/100,000 population. Thus, Mexico is positioned as the 6th country of the CKD mortality rate globally [3].

The pathophysiology of CKD involves the accumulation of uremic toxins that impair the GFR and reduce blood flow to the glomeruli. It triggers a compensatory activation of the renin–angiotensin–aldosterone system (RAAS), which leads to increased levels of angiotensin II (Ang II). Ang II causes vasoconstriction by binding to the angiotensin II receptor type 1 (AT_1_R), which is over-expressed in CKD [4,5]. Excessive RAAS stimulation results in systemic arterial hypertension (SAH), oxidative stress, inflammation, and the progressive loss of renal function [6,7]. Moreover, peripheral vasodilation decreases the production of nitric oxide (NO) because of the CKD-induced downregulation of the mitochondrial assembly protein 1 (Mas1) receptor as well as a decrease in its natural ligand concentration, angiotensin (1–7) [Ang(1–7)] [8,9]. The processes above merged into endothelial dysfunction, which plays a crucial role in CKD progression, cardiovascular complications, and mortality [9]. Therefore, it is essential to develop novel therapeutic strategies to delay CKD complications and improve patients’ life quality.

A novel strategy involves using the blue economy, defined as using sustainable ocean resources to enhance human well-being and social equity while significantly mitigating environmental risks and ecological scarcities. Mexico is a country boasting some of the world’s highest biodiversity. The Pacific Ocean Coast, a vast and uniquely warm water source, has facilitated the isolation of nearly 21 microalgae species from the genera *Phanocapsa*, *Komvophoron*, and *Phormidium*, among others. Thus, it has allowed for an opportunity for biotechnology to generate green solutions to public health problems through basic science, innovation, and technology development for the production, purification, and formulation of biotherapeutics from marine microorganisms, such as phycobiliproteins, exopolysaccharides, and other secondary metabolites [10].

Phycobiliproteins, particularly phycocyanin, have been extensively studied. However, other phycobiliproteins, including allophycocyanin and phycoerythrin, remain less explored. This study focuses on understanding the biotherapeutic action mechanism of C-phycoerythrin (CPE). CPE is a protein weighing 240–260 kDa that is composed of two monomers (α and β), which form a heterodimer assembled into a hexamer [(αβ)_3_]_2_. Notably, CPE owes its red color to the prosthetic group phycoerythrobilin, which acts as an antioxidant due to its linear tetrapyrrole structure analog to bilirubin [11,12,13]. The biotherapeutic potential of CPE in animal models has been described as anti-inflammatory [14] and a cell protector against insults that damage the liver and kidney in murine models [15,16]. The most reductionist explanation of its cytoprotective activity is associated with the antioxidant capacity. However, some reports describe the molecular mechanism of its biotherapeutic action as the prevention of the alteration of the REDOX environment and the resolution of the endoplasmic reticulum stress [17]. Meanwhile, in a cell culture, its anticancer activity has been described through the activation of apoptosis [18]. Furthermore, CPE increases the lifespan of and stress tolerance in *Caenorhabditis elegans* as an aging model [19]. Finally, the potential of biotherapeutic CPE against complications of metabolic diseases, such as dyslipidemia, has been reported [20], but the complete mechanism is still unknown. We proposed that CPE could be an antihypertensive due to its similar structure to c-phycocyanin, which has an antihypertensive activity [21,22]. Thus, this study aims to evaluate CPE’s nephroprotective mechanism due to its antioxidant and antihypertensive activities in a heuristic CKD rat model.

## 2. Results

### 2.1. Effect of CPE on Biochemical Markers Related to CKD

Table 1 shows the biomarkers of the renal function evaluation of nephrectomized animals after five weeks of CPE treatment; nephrectomized animals developed CKD due to an elevation in blood urea nitrogen (BUN, ~57%), uric acid (~77%), and serum creatinine (~68%) and the presence of proteinuria (~1240%) respective to the sham. Furthermore, CPE administration avoids azotemia with a partial amelioration of creatinine clearance (~62% for the sham) and proteinuria (~47% for the sham).

### 2.2. Effect of CPE on CKD-Induced SAH

Figure 1 shows the effect of CPE on arterial blood pressure in NFx rats during the experiment. Nephrectomized rats developed SAH two weeks after surgery, evidenced by increased systolic blood pressure (A) and diastolic blood pressure (B), which enhances the mean arterial pressure by about 150 mmHg (C) and the heart rate (D). Meanwhile, the CPE treatment in nephrectomized rats prevented CKD-induced SAH, maintaining and regulating the mean blood pressure and its components (A–D). Also, the CPE treatment per se did not reduce either component of the mean blood pressure.

### 2.3. Effect of CPE on CKD-Induced Oxidative Stress and REDOX Environment Disturbance

Renal oxidative stress (A–C) and REDOX environment (D–F) evaluations are illustrated in Figure 2, whereas nephrectomized rats showed the highest levels of reactive oxygen species (ROS) (~232%), lipid peroxidation (~330%), and protein carbonylation (~187%); regarding the REDOX environment, GSH concentration and GSH^2^/GSSG ratio were decreased (~60% and ~69%, respectively) with an elevation of GSSG concentration (~150%). The CPE treatment prevented CKD-induced oxidative stress and partially avoided disturbance in the REDOX environment through about a ~20% reduction in the GSH^2^/GSSG ratio compared to the sham.

Figure 3 deploys the impact of CPE on nitrosative stress observed in nephrectomized rats is noteworthy. CKD typically increased the nitrite concentration and iNOS expression in the kidney compared to the sham (~315% and ~106%, respectively). However, the CPE treatment mitigated these effects, increasing only 9% of the nitrite concentration and maintaining normal iNOS.

### 2.4. Effect of CPE on CKD-Disturbances Angiotensin II Receptors (AT_1_R and AT_2_R) Expression

The effect of CKD on the expression of AT_1_R (A) and AT_2_R (B) is shown in Figure 4. CKD overexpressed AT_1_R (~26%) and down-expressed AT_2_R (~32%) compared to the sham animals. However, CPE prevented the upregulation of AT_1_R (~5% higher than the sham) and maintained the normal expression of AT_2_R.

### 2.5. Protein Expression of the Mas1/Akt1/eNOS/Pathway

Figure 5 illustrates that CKD results in a down-expression of approximately 20% in the most important signaling proteins of the Mas1/p-Akt1/p(Ser-1177)-eNOS pathway to the sham. On the other hand, CPE treatment effectively sustains the normal expression of the molecules in the pathway, except for p(Ser-473)-Akt1, which only partially prevents upregulation compared to the sham.

### 2.6. Effect of CPE on CKD-Induced Renal Damage

Concerning the proteins linked to glomerular damage (Figure 6), CKD resulted in a decrease in the expression of nephrin (A) and podocin (B). Treatment with CPE prevented the loss of nephrin and podocin in the kidney. Also, the remaining renal tissue of nephrectomized rats exhibited signs of focal segmental glomerulosclerosis, glomerular hypertrophy, and tubulointerstitial fibrosis. Notably, CPE treatment partially mitigated these disturbances in renal cytoarchitecture.

## 3. Discussion

The blue economy is a potential source of alternative avenues for developing new biotherapeutics to treat non-communicable diseases, such as CKD. For instance, metabolites purified from red algae are harnessed to create new pharmaceuticals, leveraging biotechnology as a sustainable industry. Prior research has established that phycobiliproteins serve as nephroprotective molecules against acute kidney injury. They achieve this by preventing oxidative stress, disturbances in the REDOX environment, endoplasmic reticulum stress, and cellular damage [23,24,25]. Among phycobiliproteins, the two most purified forms, CPE and C-phycocyanin (CPC), exert a nephroprotective operating through remarkably similar mechanisms [14,23,24]. Despite both phycobiliproteins being nephroprotective, the biggest challenge for developing them as new biotherapeutics are the difficulties of employing them in large-scale production. This issue can be addressed using CPE, which offers a higher yield and a more straightforward purification process. For instance, the *Phormidium* genus has an advantage because it yields 42% of CPE in terms of cost-effective saline media. Moreover, the purification process for CPE is significantly more affordable than CPC from *Arthrospira* [14,26].

This study represents the first evaluation of specific mechanisms by which CPE functions to delay the onset of cardiovascular complications associated with CKD and renal damage. The 5/6 nephrectomy is a heuristic model that simulates the physiopathology of CKD, which is characterized by increased serum levels of nitrogen-containing compounds, proteinuria, and cardiovascular complications related to endothelial dysfunction, SAH, and left ventricular hypertrophy [7,27,28]. Since CKD is lately diagnosed, the importance of adjuvants in its treatment highlights CPE as a promising candidate. However, safe and toleration studies of CPE are necessary before being used as a human treatment as well as the exploration of the whole pharmacology potential of CPE in a CKD-induced HAS model.

We propose that the pharmacological mechanism action of CPE as antihypertensive is related to its hydrolysis due to gastric pH and proteases along the digestive tract, resulting in the release of vasoactive chromo-peptides and PEB, which are subsequently absorbed in the small intestine, as occurs with CPC [29]. Thus, PEB could be responsible for CPE’s biological effects because PEB is a near-identical chemical composition to phycocyanobilin (PCB), the active metabolite of phycocyanin [23,25]. PEB and PCB are both isomers where either ethylene or ethyl group is attached at C18, respectively. The other difference between them is the unsaturation between C15 and C16. In addition, both scavenged ROS and other free radicals have the same antioxidant potential [13,24]. However, further research is required to evaluate the products of CPE in simulated in vitro gastrointestinal digestion to elucidate its bioactive molecules.

In CKD, the overactivation of RAAS, in conjunction with oxidative stress and uremic state, contributes to vascular damage by activating AT_1_R-stimulating protein phosphatase 2A, resulting in eNOS dephosphorylation and a low level of NO. This process increases the peripherical vascular resistance, which is not compensated by AT_2_R and Mas1 pathways [9,26]. Additionally, the damage-associated molecule patterns (DAMS) activate Toll-like receptor 4 (TLR4), amplifying inflammation, oxidative stress, cellular damage, and endothelial dysfunction [30].

Our results demonstrate that CPE prevents endothelial dysfunction by normalizing the vasoactive/vasoconstrictive state in the kidney, modulating the expression of AT_1_R and AT_2_R/Mas1 and allowing for NO production. CPE also prevents the downregulation of the Mas1/AKT1/eNOS pathway, enhancing the vasorelaxant effect through NO production and avoiding endothelial dysfunction in arteriole in the remanent functional kidney. The daily CPE treatment prevents endothelial dysfunction and the development of cardiovascular complications, such as SAH, reducing oxidative stress and proteinuria [31] by preserving nephrin in the podocyte diaphragm, as observed with other renal pathologies [17]. Also, the CPE has an antihypertensive effect through its anti-inflammatory properties that maintains the regular expression of AT_2_R and Mas1, which are quite often reported by reducing immune cell infiltration in the endothelium and decreasing pro-inflammatory cytokines in the glomerular epithelium [32,33,34], and its metabolite (PEB) which is a homolog of bilirubin, activates the biliverdin reductase, improving these effects [35]. All these mechanisms merge to protect the remanent functional kidney in CKD and delay the progression of its cardiovascular complications. The development of SAH secondary to CKD contributes to the depletion of the reduced GFR, causing the accumulation of nitrogenous compounds, which contributes to increased osmolarity and inflammation [6,36]. This process is prevented by CPE treatment.

In summary, Figure 7 shows the possible mechanism of CPE as an antihypertensive and nephroprotector in treating CKD. CPE modulates glomerular arterioles’ vasodilation/vasoconstriction state by maintaining regular AT_1_R, AT_2_R, and Mas1 expression.

## 4. Materials and Methods

### 4.1. CPE Obtention

The CPE was obtained from *Phormidium persicinum,* as previously reported, with minor modification [17]. Briefly, *P. persicinum* was cultured in an NM medium. Incubation was carried out at a constant temperature (21 ± 2 °C), with aeration (provided by an air pump), and under green LED illumination (24W, 3000 Lx) on a 12 h photoperiod (lights on at 08:00 AM). The biomass was separated by centrifugation at 10,000× *g* for 1 min, and 5 g of cell pellet was resuspended in 20 mL of 10 mM phosphate buffer at pH 7.4 (PB). Three freeze–thaw cycles at −20 °C were performed for cell disruption. The protean extract was centrifugated four times at 21,400× *g* for 10 min at 4 °C. CPE was separated using a 33 × 4.7 cm column containing a pre-calibrated Sephadex G-50 with 10 mM phosphate buffer solution at pH 7.4 (PBS). In darkness, the pink fractions were precipitated with a saturated (NH_4_)_2_SO_4_ solution at 0 °C for 24 h. Subsequently, CPE was dialyzed with nitrocellulose membranes, and CPE concentration was quantified using a standard curve. The purified CPE was solubilized in sucrose 5 mM and frozen at −20 °C to await rat administration.

### 4.2. Animals

Twenty normotensives male Wistar rats weighing 250–300 g were acclimatized for two weeks before the experiment. During this period, the animals were kept in a room at a constant temperature of 21 ± 2 °C, with 60% relative humidity, and a 12 h photoperiod starting at 08:00. The animals had ad libitum access to standard food and tap water. All experimental procedures followed the regulation outlined in NOM-062-ZOO-1999 and were approved by the institutional committee (CEI-ENCB) for the care and use of laboratory animals with the approval code ZOO-003-2022 [37].

Two weeks before surgery, the animals underwent adaptation to a non-invasive blood pressure evaluation system CODA^®^ (Kent Scientific, Torrington, CT, USA). As previously reported, normotensive animals, characterized by arterial blood pressure under 120/80 mmHg, were selected for the CKD model [22].

The animals were randomly assigned to four groups (n = 5): (1) sham + vehicle (PBS); (2) sham + C-phycoerythrin (100 mg/kg, CPE); (3) 5/6 nephrectomy (NFx) + vehicle; and (4) NFx + CPE. CPE and vehicle were administrated daily by oral gavage one week after surgery and continued for five weeks.

At the end of the treatment, 24 h urine was collected for renal function evaluation. Subsequently, animals were euthanized with sodium pentobarbital overdose [130 mg/kg, intraperitoneally). Blood was obtained by cardiac puncture under deep anesthesia, and serum was separated and stored at −80 °C in an ultra-low-temperature freezer. Kidneys were excised, with one half immediately stored at −80 °C, and the remaining were soaked in 10% formalin for 24 h. All samples were preserved until molecular or biochemical evaluations.

### 4.3. 5/6 NFx-Induced CKD Model

Briefly, rats were anesthetized with a single intraperitoneal dose of sodium pentobarbital (35 mg/kg). Under aseptic conditions, a ventral laparotomy was conducted to expose the left kidney. Two of the three branches were occluded with 4-0 black silk sutures, resulting in visible infarction in 2/3 of the kidney. Subsequently, a nephrectomy of the right kidney was performed by occluding renal blood vessels and the ureter. Following the surgery, animals received tramadol (10 mg/kg orally) and enrofloxacin (10 mg/kg subcutaneously) for three days to alleviate pain and prevent potential infections [22,37].

### 4.4. Cardiovascular Evaluation

Systolic and diastolic blood pressure (SBP and DBP, respectively), mean arterial pressure (MAP), and heart rate (HR) were measured in the tail twice a week utilizing the non-invasive system CODA^®^ (Kent Scientific, Torrington, CT, USA)

### 4.5. Renal Function Analysis

At the completion of CPE treatment, animals were individually housed in Tecniplast™ metabolic cages (Metabolic Cage Systems for Rodents, ThermoFisher Scientific, Waltham, MA, USA) to collect 24 h urine. Intracardiac blood samples were obtained during euthanasia under deep anesthesia, and serum was subsequently separated by centrifugation at 3500× *g* for 5 min. Blood urea nitrogen (BUN), serum uric acid, creatinine, urine creatinine, and proteins were quantified using SPINREACT^®^ (Girona, Spain) kits.

### 4.6. Oxidative Stress and REDOX Environment Markers

The oxidative stress and REDOX environment markers were evaluated as previously described with mild modification [22]. Briefly, the kidneys were homogenized in 3 mL of PBS. Protein quantification was performed using 1 μL of homogenate mixed with 99 μL of deionized water and 900 µL of Bradford reagent. The mixture was incubated at room temperature for 10 min, and absorbance was measured at 595 nm.

For the quantification of reactive oxygen species (ROS), 10 μL of homogenized kidney was combined with 1940 μL of TRIS:HEPES (18:1, *v*/*v*) at pH 7.4 and 50 μL of 2′,7′-dichlorofluorescine diacetate (DCFH-DA, 0.2 mg/mL in methanol). The mixture was then incubated in a water bath at 37 °C for one hour. The reaction was halted by a thermal shock in an ice bath, and the fluorescence was measured at 488 nm of excitation and 525 nm of emission. ROS content was calculated using a 2′,7′-dichlorofluorescein (2′,7′-DCF) standard curve. The results were expressed as ng 2′,7′-DCF formed per mg of protein per hour.

The lipid peroxidation assay was performed by adding 500 μL of the homogenized kidney to 500 μL of PBS and mixing it with 8 mL of methanol/chloroform (2:1). The mixture was vortexed for 30 s and kept at 4 °C for 30 min in darkness. Subsequently, the aqueous phase was removed, and 2 mL of the organic phase was utilized for fluorescence measurement at 370 nm of excitation and 430 nm of emission using a fluorometer adjusted to 140 fluorescence units with 1 μg/mL of quinine in 0.05 M H_2_SO_4_. The outcomes were expressed as relative fluorescence units (RFU) per mg of protein.

To quantify protein carbonylation (PC), 5 μL of homogenized kidney with 250 μL 2,4-dinitrophenylhydrazine 10 mM (solubilized in 2.5 M HCl, DNPH) and incubated for one hour in darkness. After centrifugation at 1200× *g* for 10 min, the supernatant was discarded, and the pellet was resuspended in 250 μL of 20% trichloroacetic acid. This process was repeated twice, and the washed pellet was then resuspended in 500 μL of 6 M guanidine chloride. Following this, 100 μL of the suspension was loaded in a 96-well plate and incubated for 10 min at 37 °C; the absorbance was measured in the supernatant at 370 nm, and PC was calculated using the molar absorption coefficient of 22,000 M^−1^cm^−1^ relative to protein concentration. The results were expressed as nmol of DNPH per mg of protein.

The REDOX environment evaluation included quantifying reduced glutathione (GSH) and oxidized glutathione (GSSG). In this process, 500 μL of the homogenized kidney was mixed with 100 μL of 30% phosphoric acid. After centrifugation at 19,000× *g* for 30 min at 4 °C, 30 μL of the supernatant was added to 1870 μL of FEDTA (PBS with 5 mM EDTA at pH 8.0) for GSH quantification. For GSSG quantification, 100 μL of supernatant was mixed with 50 μL of N-ethylmaleimide (5 mg/mL solubilized in deionized water) and added to 1.7 mL of 0.1 N NaOH. Finally, 100 μL of o-phthaldialdehyde (1 mg/mL solubilized in methanol) was added to evaluate fluorescence (350 nm for excitation and 420 nm for emission) after 10 min of incubation, and GSH or GSSG content was calculated using a GSH and GSSG standard curve, respectively. The results were expressed as ng of GSH or GSSG per milligram of protein, and the GSSG^2^/GSH ratio was calculated.

Nitrites (NO_2_) were determined by adding 100 μL of the homogenized kidney to 300 μL of concentrated HCl, followed by 250 μL of 20% metallic zinc suspension. The mixture was incubated at 37 °C for one hour, then centrifuged at 4000× *g* for 2 min. Subsequently, 50 μL of the supernatant was loaded into a 96-well plate with 100 μL of Griess reagent. After incubating for 15 min at room temperature, the nitrite content in the supernatant was measured at 530 nm of absorbance using a standard curve with NaNO_2_. Results were expressed as μg of NO_2_^−^ per mg of protein.

### 4.7. Western Blotting

100 μL of the homogenate (n = 3 per group) was combined with 100 μL of a complete protease inhibitor cocktail (Santa Cruz Biotechnology) and subjected to a 3 min incubation in a boiling water bath. Electrophoresis was carried out by loading 50 µg protein samples per lane onto a 10 or 15% SDS-PAGE gel and separating them at a constant voltage of 100 V for one hour. The proteins were then electrotransferred to PVDF membranes (Millipore, Bedford, MA, USA) using a Trans-Blot Turbo System (Bio-Rad, Hercules, CA, USA) at 25 V and 2.5 A for 10 min. Subsequently, membranes were blocked with PBST (PBS with 0.05% Tween 20 and 5% low-fat MilkSvelty^®^) for 1 h under constant agitation at room temperature. The membranes were incubated overnight at 4 °C in PBST with primary antibodies (Santa Cruz Biotechnology, Dallas, TX, USA) diluted 1:1000 for Mas1 (sc-390453), p(Ser-1177)-eNOS (sc-81510), eNOS (sc-376751), iNOS (sc-7271), and AT_1_R (sc-515884) and 1:500 for AT_2_R (sc-518054), nephrin (sc-376522), podocin (sc-518088), and p(Ser-473)-AKT1 (sc-293125).

Following incubation, membranes were washed three times with fresh PBST (30 min per wash) and then incubated for one hour under constant agitation at room temperature, with a specific secondary antibody linked to HPR (Santa Cruz Biotechnology, Dallas, TX, USA) diluted 1:3000. Subsequently, it underwent three additional washes with fresh PBST (30 min per wash). Finally, protein bands were revealed using chemiluminescence on photographic plates using Luminata™ Forte^®^ (Millipore, Billerica, MA, USA). β-actin (Santa Cruz Biotechnology, Dallas, TX, USA; sc-47778, diluted: 1:1500) was used as constitutive protein expression and loading control. The optical density (O.D.) from protein bands was analyzed with ImageJ/FIJI (1.46v. NIH, Bethesda, MD, USA), and the results were presented as the O.D. of protein/O.D. of β-actin ratio.

### 4.8. Histopathologic Stain

For the histopathology stain, kidneys were fixed in 10% neutral buffered paraformaldehyde for 24 h, then embedded in paraffin according to standard protocol. Serial sections of 5 μm thickness were cut and stained with the standard hematoxylin-eosin (HE) technique. Samples were examined using optical microscopy by an observer blind to the experiment.

### 4.9. Statistical Analysis

All data are expressed as the mean ± standard error. The variables evaluated (serum renal function markers, oxidative stress markers, and protein expression by Western blot) were analyzed by two-way analysis of variance (ANOVA) using CKD and CPE treatment as factors, while hemodynamic variables were analyzed by two-way repeated measures ANOVA using time and group as factors. All ANOVAS were followed by the Tukey post hoc test using GraphPad Prism 8 (GraphPad Software, Boston, MA, USA). Statistical significance was considered as *p* < 0.05.

## 5. Conclusions

The present study demonstrates the renal mechanisms associated with the antihypertensive effect of CPE through the modulation of the vasoconstrictive/vasodilative state of AT_1_R, AT_2_R, and the Mas1/Akt/eNOS pathway and the reduction in oxidative stress with the protection of the remanent functional kidney. This is a sustainable novel strategy to treat CKD according to the blue economy, which improves human well-being and social equity while significantly mitigating environmental risks and ecological scarcities.

## Figures and Tables

**Figure 1 marinedrugs-22-00337-f001:**
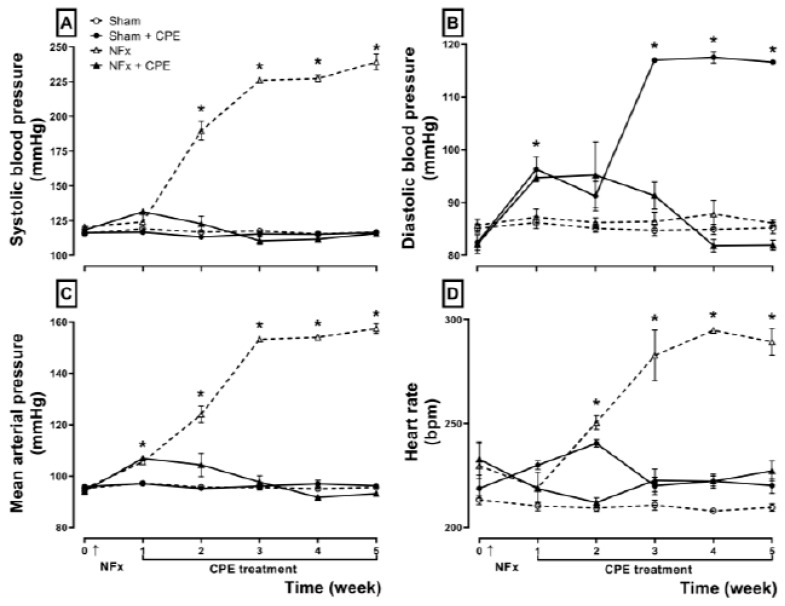
The effect of CPE treatment on CKD-induced systemic arterial hypertension. Systolic blood pressure (SBP, **A**), diastolic blood pressure (DBP, **B**), mean arterial pressure (MAP, **C**), and heart rate (HR, **D**). The arrow represents the NFx surgery. The values represent mean ± SEM. (*) *p* < 0.05 compared to the sham at the same time. RM two-way ANOVA and SNK post hoc test.

**Figure 2 marinedrugs-22-00337-f002:**
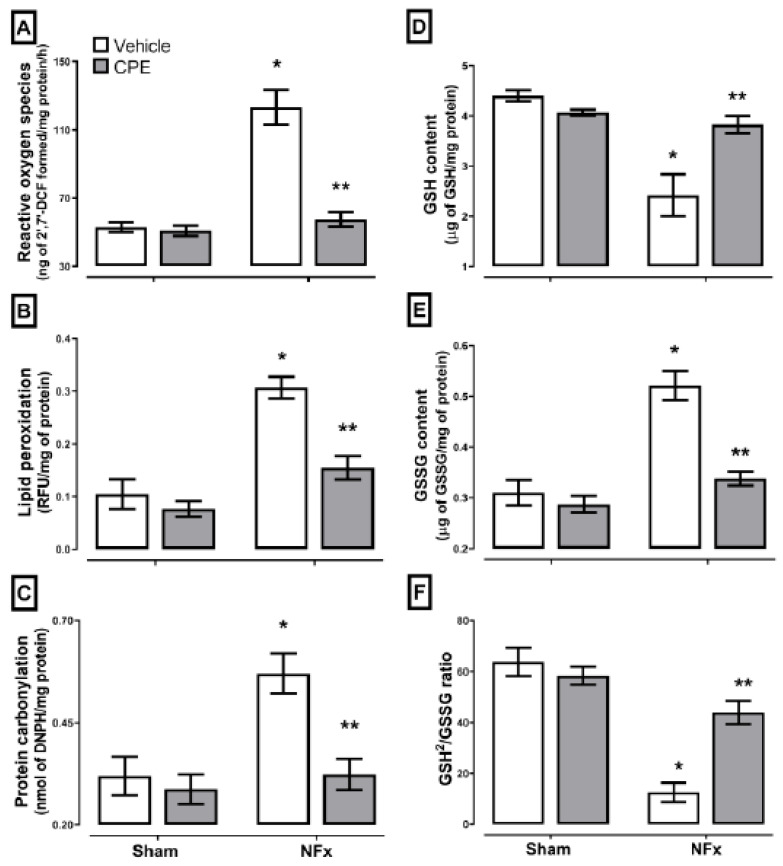
The effect of CPE administration on CKD-induced oxidative stress and REDOX environment alteration in kidney. ROS (**A**), lipid peroxidation (**B**), and protein carbonylation (**C**) are oxidative stress markers. Meanwhile, the GSH (**D**), GSSG (**E**), and GSH^2^/GSSG ratio (**F**) are REDOX environment markers. The values represent mean ± SEM. (*) *p* < 0.05 compared to the sham + vehicle. (**) *p* < 0.05 compared to the NFx + vehicle. Two-way ANOVA and SNK post hoc test.

**Figure 3 marinedrugs-22-00337-f003:**
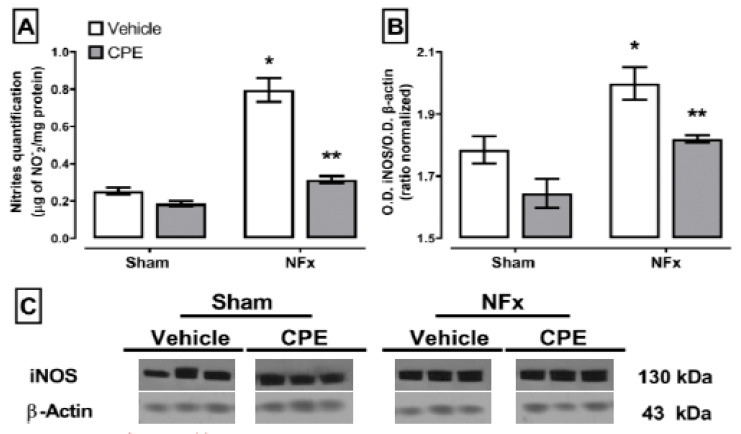
The effect of CPE administration on CKD-induced nitrosative stress in kidney. Nitrites quantification (**A**) and iNOS expression (**B**). (**C**) shows representative blot of protein expression in kidney The values represent mean ± SEM. (*) *p* < 0.05 compared to the sham + vehicle. (**) *p* < 0.05 compared to the NFx + vehicle. Two-way ANOVA and SNK post hoc test.

**Figure 4 marinedrugs-22-00337-f004:**
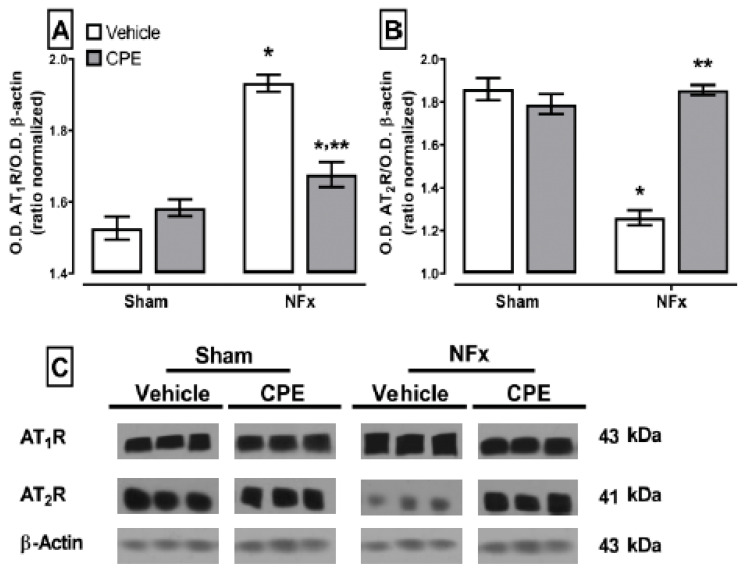
The effect of CPE administration on the CKD-induced disturbance expression of angiotensin receptors (AT_1_R (**A**), AT_2_R (**B**)). (**C**) The blot of angiotensin receptors’ expression in the kidney. The values represent mean ± SEM. (*) *p* < 0.05 compared to the sham + vehicle. (**) *p* < 0.05 compared to the NFx + vehicle. Two-way ANOVA and SNK post hoc test.

**Figure 5 marinedrugs-22-00337-f005:**
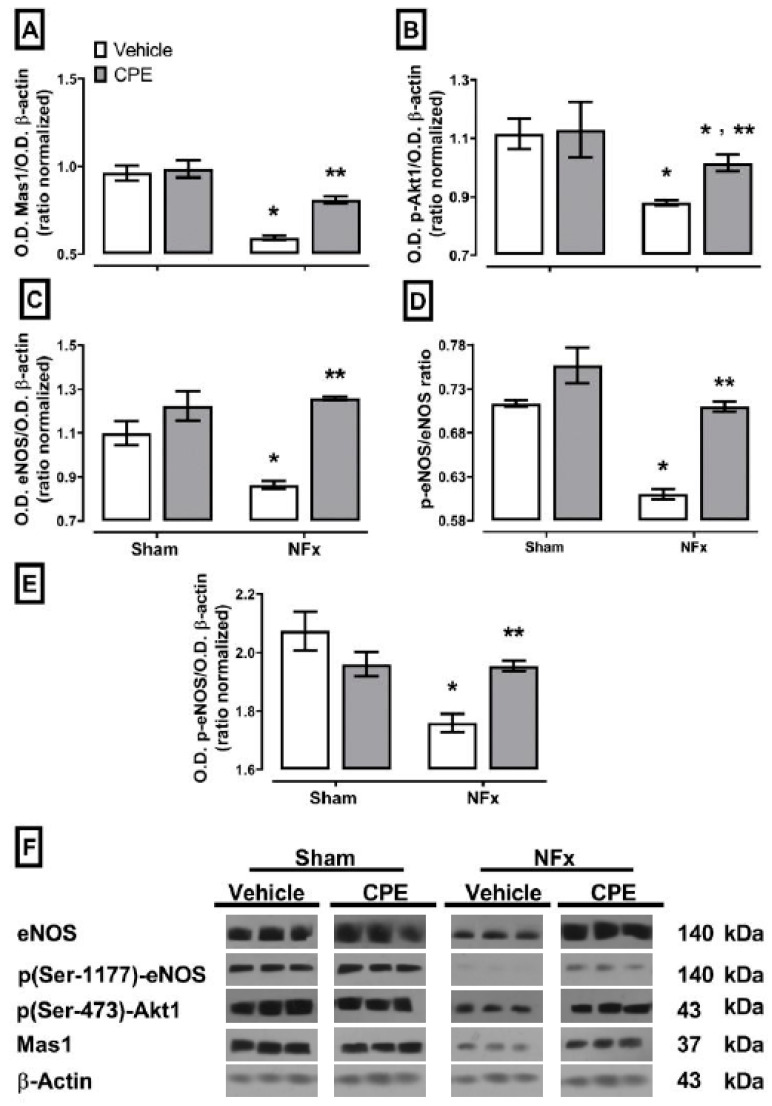
The effect of CPE administration on the CKD-induced suppression of the Mas1 (**A**)-p(Ser-473)-Akt1 (**B**)-p(Ser-1177)-eNOS (**C**) signaling pathway as well as eNOS (**D**) and the p(Ser-1177)-eNOS/eNOS ratio (**E**). (**F**) The blot of protein expression in the kidney of NFx rats. The values represent mean ± SEM. (*) *p* < 0.05 compared to the sham + vehicle. (**) *p* < 0.05 compared to the NFx + vehicle. Two-way ANOVA and SNK post hoc test.

**Figure 6 marinedrugs-22-00337-f006:**
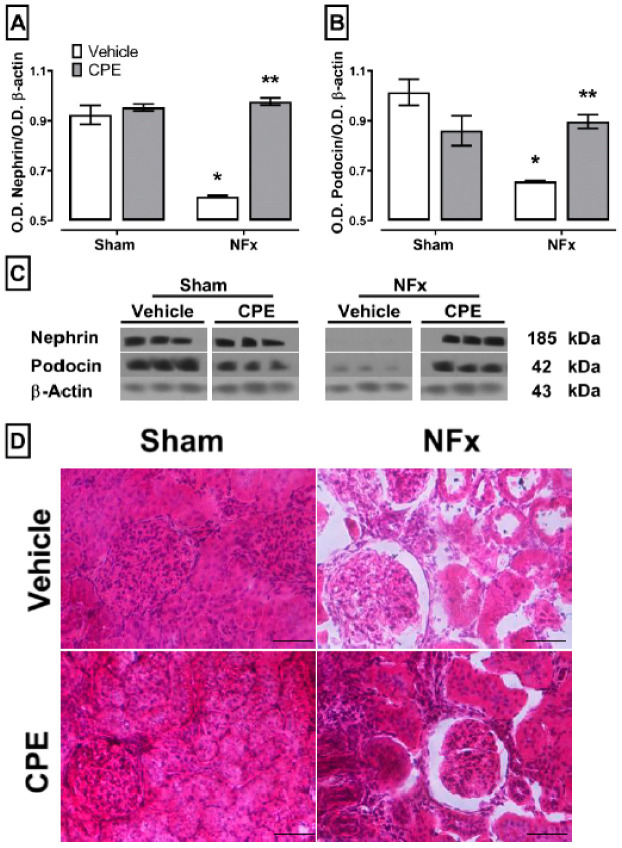
The effect of CPE on the CKD-induced down regulation of nephrin (**A**) and podocin (**B**) and kidney damage. (**C**) The blot of protein expression in the kidney. (**D**) Representative photomicrographs of the renal cortex stained with hematoxylin and eosin. The lower right bar of photomicrographs represents 250 µm. The values represent mean ± SEM. (*) *p* < 0.05 compared to the sham + vehicle. (**) *p* < 0.05 compared to the NFx + vehicle. Two-way ANOVA and SNK post hoc test.

**Figure 7 marinedrugs-22-00337-f007:**
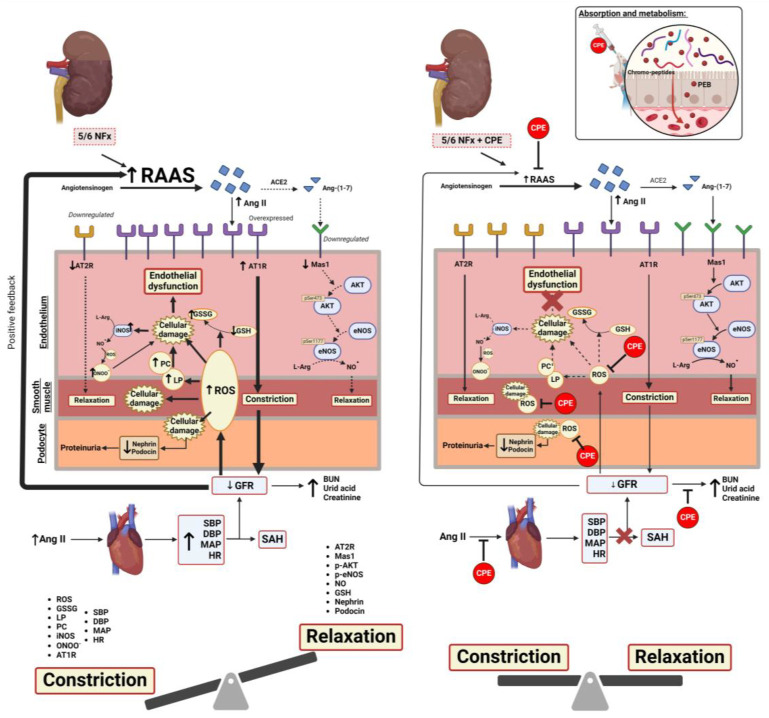
The proposed model for the protective mechanism of CPE against CKD-induced HAS. Left: 5/6 NFx induces RAAS overactivation, leading to an increase in Ang II. This results in the upregulation of AT_1_R and the downregulation of AT_2_R and Mas1. These alterations establish a vasopressive state, reducing the GFR and allowing for the accumulation of uremic compounds. The sustained production of ROS, LP, and PC surpasses endogenous antioxidant systems, causing oxidative stress-mediated cellular damage and inflammation in the endothelium and smooth vascular muscle of the arteriole. This inflammatory environment triggers the expression of iNOS and the further oxidation of NO^·^ to ONOO^−^, contributing to nitrative stress. Simultaneously, cellular damage leads to depletion in the glomeruli filtering barrier by diminishing nephrin and podocin, allowing for the occurrence of proteinuria. The persistent overactivation of RAAS increases the peripheral swiftness of arteries, culminating in endothelial dysfunction and the establishment of SAH. This perpetuates the decline in the GFR and the progression of CKD. Upon oral-gavage administration, CPE undergoes digestion, releasing chromo-peptides and PEB. These components mitigate ROS-mediated cellular damage through scavenging. Additionally, CPE prevents dysregulation in vasomodulation by normalizing the expression of AT_1_R, AT_2_R, and the Mas1/Akt1/eNOS pathway. Through these mechanisms, CPE maintains kidney function and prevents the establishment of SAH. CKD: Chronic kidney disease; BUN: Blood ureic nitrogen; GFR: Glomerular filtration rate; RAAS: Renin angiotensin aldosterone system; SBP: Systolic blood pressure; DBP: Diastolic blood pressure; MAP: Mean arterial pressure; ROS: Reactive oxygen species; LP: Lipid peroxidation; PC: Protein carbonylation; CPE: C-phycoerythrin; PEB: Phycoerythrobilin. Dash line represents reduced pathway; bold line represents increased pathway. Created with Biorender.com.

**Table 1 marinedrugs-22-00337-t001:** Effect of CPE on renal function in nephrectomized rats.

	BUN (mg/dL)	Uric Acid (mg/dL)	Serum Creatinine (mg/dL)	Creatinine Clearance (mL/min)	Proteinuria (mg/dL)
Sham	58.61 ± 0.41	3.21 ± 0.13	0.80 ± 0.02	0.35 ± 0.08	2.03 ± 0.11
Sham + CPE	59.73 ± 1.16	3.40 ± 0.12	0.75 ± 0.02	0.52 ± 0.04	2.28 ± 0.11
NFx	83.56 ± 5.97 *	6.16 ± 0.21 *	1.28 ± 0.15 *	1.40 ± 0.13 *	5.71 ± 0.45 *
NFx + CPE	63.38 ± 3.05 **	4.45 ± 0. 10 *, **	0.96 ± 0.04 **	0.76 ± 0.07 *, **	3.60 ± 0.47 *, **

The values represent mean ± SEM. (*) *p* < 0.05 compared to the sham. (**) *p* < 0.05 compared to NFx. Two-way ANOVA and SNK post hoc test.

## Data Availability

The datasets generated during the current study are available from the corresponding author upon reasonable request.

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
