# Peer review of "C-Phycoerythrin Prevents Chronic Kidney Disease-Induced Systemic Arterial Hypertension, Avoiding Oxidative Stress and Vascular Dysfunction in Remanent Functional Kidney"

_marinedrugs, 2024, doi:10.3390/md22080337_

Round 1
Reviewer 1 Report
Comments and Suggestions for Authors
There are several major concerns in this study.
They did not confirm whether CKD model were successfully bulit.
Biochemical indicators in the blood and urea were not measured, which is critical to measure the renal funcion.
The quality of Western blotting bands was low; and they should not separate the groups.
The quality of HE staining images was low
Comments on the Quality of English LanguageThe language is ok.
Author Response
We appreciate your taking the time to revise the article. We appreciate your taking the time to revise the article. Your comments helped us greatly in amending it. We attended to all the points that the reviewers questioned us about. Below are the responses to your comments concerning:
Review Report (Reviewer 1). There are several major concerns in this study.
- They did not confirm whether the CKD model was successfully built.
Measuring biochemical parameters such as BUN, creatinine clearance, uric acid, and proteinuria confirmed the CKD model. These results are shown in Table 1 since the first draft. Also, these results demonstrated that CKD was induced by NFx 5/6, as other research groups have proposed a heuristic model to reproduce all the physiopathology of CKD [1–5].
- Biochemical indicators in the blood and urea were not measured, which is critical to measure the renal function.
As mentioned in the first question, table 1 shows these results corroborating the CKD model.
- The quality of Western blotting bands was low; and they should not separate the groups.
Thanks for your comments. We amended the figures by increasing the size of the blots and photomicrographs. Regarding the blots, these were carried out over several days, running the gel and developing each membrane for each marker. The analyst processed actin, AT1R, AT2R, eNOS, p-eNOS, and p-AKT in two weeks in the same order, while iNOS, Mas1, nephrin, and podocin were evaluated in the next two weeks with another order. Thus, just to place the sample blots into the figures, we decided to cut them. However, each blot was analyzed without any editing. Additionally, we know that this technique, as we are processing it, is semiquantitative, but using Image J for image analysis, we selected the same area for all bands of each marker, thereby reducing the error of changes in intensity due to human error. Even having the control groups without a significant difference and the difference between the groups with CKD, as observed in other reports, make us think the analysis is valid. If, despite that, it is necessary to repeat any of the blots, we would need more time.
- The quality of HE staining images was low
We amended it.
Please see the attachment for the references.

Reviewer 2 Report
Comments and Suggestions for Authors
Authors investigated and demonstrated that C-phycoerythrin (CPE) could prevent hypertension, oxidative stress and vascular dysfuntion observed in Chronic Kidney Disease produced in rats by 5/6 nephrectomy. A mechanism of CPE action was also proposed.
Experimental procedures were according with the aim and the evaluated parameters were adequated.
I proposed that the manuscript is suitable for publication
Author Response
We appreciate your taking the time to revise the article. We appreciate your taking the time to revise the article. Your comments helped us greatly in amending it. We attended to all the points that the reviewers questioned us about. Below are the responses to your comments concerning:
Review Report (Reviewer 2)
- Authors investigated and demonstrated that C-phycoerythrin (CPE) could prevent hypertension, oxidative stress and vascular dysfuntion observed in Chronic Kidney Disease produced in rats by 5/6 nephrectomy. A mechanism of CPE action was also proposed.
- Experimental procedures were according with the aim and the evaluated parameters were adequated.
- I proposed that the manuscript is suitable for publication
Thanks for your comments.
Reviewer 3 Report
Comments and Suggestions for Authors
This original article is about the protection of phycoerythrin in a model of renal damage induced by 5/6 nephrectomy in rats. The experiments were well conducted; however, part of their methodology needs to be clarified. The observations are mentioned below.
1. This original work is about C-PE. However, very little is said about her. Please add more information about C-PE.
2. Please verify the total number of rats used in the study.
3. Did the sham groups also receive tramadol and enrofloxacin? Would it modify the results in the other groups?
4. It is essential to mention how they obtained phycoerythrin.
5. For the determination of oxidative stress markers and REDOX markers. Were standard curves used to interpolate the measurements?
6. Although it is mentioned in the tables and figures, it is necessary to mention the statistical analysis used, the result presented in the graphs (MEAN +- SEM, etc.), and the program used for statistics and figures in the Methodology section.
7. Homogenize the CPE abbreviation.
8. In humans, what dose would be required?
9. Please check the online GSH2/GSSG ratio108.
10. It may be advisable to change the figure of the WB p(SER-1177)-eNOS for the vehicle group.
11. Figure 7. It is unclear why the endothelium is considered the place where the modification of cell signaling occurs, nor the origin of the increase in ROS.
12. Please mention the positive and negative controls used in each test.
Author Response
We appreciate your taking the time to revise the article. We appreciate your taking the time to revise the article. Your comments helped us greatly in amending it. We attended to all the points that the reviewers questioned us about. Below are the responses to your comments concerning:
- This original work is about C-PE. However, very little is said about her. Please add more information about C-PE.
We amended it. We added more information about CPE and its beneficial activities between the lines 69-89 (marked in yellow)
- Please verify the total number of rats used in the study.
Thanks for your comments; we amended it. We used 20 rats divided into four groups, each with five animals (line 283, marked in yellow)
- Did the sham groups also receive tramadol and enrofloxacin? Would it modify the results in the other groups?
Thanks for the question. The sham groups were administered tramadol and enrofloxacin at the same dose as the NFx groups. Since the administration lasted three days, the effects of the drugs did not modify any measure at the end of week five, as shown in Table 1. Also, in other studies, the procedure was mandatory and did not affect the results [6–8]. However, as the sham animals received the surgical procedure, the treatment was for avoiding pain and bacterial infection, and this procedure is mandatory by the Ethical Committee of ENCB-IPN
- It is essential to mention how they obtained phycoerythrin.
Thanks for your comments. We added the CPE purification in lines 258-271 (marked in yellow).
- For the determination of oxidative stress markers and REDOX markers. Were standard curves used to interpolate the measurements?
The fluorescence or absorbance for the ROS, GSH, GSSG, and nitrites tests was interpolated in a standard curve. Meanwhile, the PC employed a molar absorption coefficient. We added that information in the methodology section of each test.
- Although it is mentioned in the tables and figures, it is necessary to mention the statistical analysis used, the result presented in the graphs (MEAN +- SEM, etc.), and the program used for statistics and figures in the Methodology section.
Thanks for your comments; we amended it in the last section of methodology 4.9, lines 395-401 (marked in yellow).
- Homogenize the CPE abbreviation.
We amended it
- In humans, what dose would be required?
This study did not focus on translational application but on the antihypertensive mechanism of CPE. However, the human equivalent dose (HED) could be estimated as [9]:
HED (mg/kg) = Animal dose (mg/kg) ⨯ 0.162
HED = 100 (mg/kg) ⨯ 0.162 = 617.28 mg/kg
- Please check the online GSH2/GSSG ratio108.
Thanks for your comments; it was amended (marked in yellow).
- It may be advisable to change the figure of the WB p(SER-1177)-eNOS for the vehicle group.
In the case of the vehicle group, the relative expression in panel D in sham and sham + CPE did not statistical difference. However, we supposed that phosphorylated eNOS tends to increase in sham + CPE, promoting vasorelaxation and avoiding endothelial dysfunction. In the blot, the protein levels are well represented, so we believe that this result enhances the hypothesis of the antihypertensive action of CPE
- Figure 7. It is unclear why the endothelium is considered the place where the modification of cell signaling occurs, nor the origin of the increase in ROS.
In CKD, ROS production in the kidney occurs in all cells, endothelium, smooth muscle, and all renal cells [10–12]. We amended Figure 7, we tried to evidence the ROS formation in all cells. However, we are in accordance with other research groups that propose that endothelial dysfunction is pivotal for promoting inflammation and remodeling processes in vascular and renal tissues where ROS production triggers all these processes [13,14]
- Please mention the positive and negative controls used in each test.
The NFx group was established as a positive control of CKD due to 5/6 NFx reproducing CKD; meanwhile, the sham with the vehicle was the negative control because this group received the surgery procedure with drug treatment without the nephrectomy process.
Please see the attachment for the references.

Reviewer 4 Report
Comments and Suggestions for Authors
The manuscript presents an in-depth investigation into the antihypertensive and nephroprotective effects of C-phycoerythrin (CPE) in a rat model of chronic kidney disease (CKD). The study is well-organized, providing a clear rationale for the use of CPE as a potential therapeutic agent to mitigate CKD-induced systemic arterial hypertension (SAH) and vascular dysfunction. Although interesting, there are several issues need to be addressed before it is considered for publication in the Journal of Personalized Medicine. Although interesting, there are several issues need to be addressed before it is considered for publication in the journal of Marine Drugs
(1)While the study employs 1H-NMR metabolomics, it does not provide a thorough justification for choosing this method over mass spectrometry-based metabolomics, which is known for its higher sensitivity and specificity.
(2)The manuscript could benefit from a more detailed discussion on the clinical implications of the findings. While the study provides valuable insights into the potential of CPE as a therapeutic agent, it lacks a comprehensive analysis of how these findings could be translated into clinical practice and what further research is needed to move towards clinical applications.
(3)The use of a single animal model (5/6 nephrectomy-induced CKD in Wistar rats) may not fully capture the complexity and variability of CKD in humans. Including different models or discussing the limitations of the chosen model would provide a more balanced perspective on the study’s applicability.
(4)The duration of the treatment and observation period (five weeks) may be insufficient to fully understand the long-term effects and safety of CPE treatment. Longer studies are needed to assess the chronic impacts and potential side effects of CPE.
(5)The manuscript lacks detailed information on the dosage of CPE used in the treatment groups. Providing a clear rationale for the chosen dosage, along with information on dose-response relationships, is crucial for understanding the therapeutic potential and safety profile of CPE. This information is also necessary for translating the findings to clinical settings.
Comments on the Quality of English LanguageI recommend a thorough review to ensure consistency and precision in language.
Author Response
We appreciate your taking the time to revise the article. We appreciate your taking the time to revise the article. Your comments helped us greatly in amending it. We attended to all the points that the reviewers questioned us about. Below are the responses to your comments concerning:
- While the study employs 1H-NMR metabolomics, it does not provide a thorough justification for choosing this method over mass spectrometry-based metabolomics, which is known for its higher sensitivity and specificity.
We did not employ 1H-NMR metabolomics nor mass spectrometry-based metabolomics.
- The manuscript could benefit from a more detailed discussion of the clinical implications of the findings. While the study provides valuable insights into the potential of CPE as a therapeutic agent, it lacks a comprehensive analysis of how these findings could be translated into clinical practice and what further research is needed to move towards clinical applications.
This study did not focus on the translational application because there are a few studies about the security of CPE; for example, the unique research about the sub-chronic treatment employed lower doses than the therapeutics we used [15]. These results are encouraging, but we should take them with caution for translational studies until complete toxicological studies are completed
- The use of a single animal model (5/6 nephrectomy-induced CKD in Wistar rats) may not fully capture the complexity and variability of CKD in humans. Including different models or discussing the limitations of the chosen model would provide a more balanced perspective on the study’s applicability.
The 5/6 nephrectomy is a heuristic model that reproduces the pathophysiology of CKD in humans, specifically, the development and progression of systemic arterial hypertension. This study was carried out in accordance with other research groups that employ this model for the study of CKD in rats. For this reason, we did not consider this model limited in terms of applicability [2–6,16–19].
- The duration of the treatment and observation period (five weeks) may be insufficient to fully understand the long-term effects and safety of CPE treatment. Longer studies are needed to assess the chronic impacts and potential side effects of CPE, nevertheless
The use of CPE as a treatment for diseases is poorly studied, and there is not enough information about its therapeutic mechanisms. This question is assertive, but the aim of this study was to increase knowledge about CPE's therapeutic mechanisms as an antihypertensive in CKD. Moreover, this paper does not aim to conduct complete safety studies. We are conducting these respective studies, and in six months, we will finish, but until now, the preliminary results are promising
- The manuscript lacks detailed information on the dosage of CPE used in the treatment groups. Providing a clear rationale for the chosen dosage and information on dose-response relationships is crucial for understanding the therapeutic potential and safety profile of CPE. This information is also necessary for translating the findings to clinical settings.
This question is widely assertive. Our research group and others have worked from protein extracts rich in phycobiliproteins to pure phycobiliproteins such as phycocyanin and phycoerythrin. In different animal models that reproduce kidney and metabolic diseases, as well as models with damage to several organs, the doses of phycobiliproteins, phycocyanin, and phycoerythrin have been selected from dose-response curves, in which the dose with the best effect is 100 mg/Kg/d [6,8,20–24]. In particular, CPE from P. persicinum was selected from a previous study on acute kidney injury [23,24].
Please see the attachment for the references.

Round 2
Reviewer 1 Report
Comments and Suggestions for Authors
The quality of data is low.
Author Response
Comment 1: The quality of the data is low.Response: Thank you for your observation. We have taken steps to enhance the quality of the data. Please provide any further comments you may have at your earliest convenience, and suggestions on how we can improve our paper would be greatly appreciated. Thank you.

Round 3
Reviewer 1 Report
Comments and Suggestions for Authors
It is acceptable.